# Evaluation of water conservation function in the Xiongan New Area based on the comprehensive index method

Yanling Chen[1,2,3], Adu Gong[1,2,3]*, Tingting Zeng[1,2,3], Yuqing Yang[1,2,3]

**1** Beijing Key Laboratory of Environmental Remote Sensing and Digital City, Beijing Normal University, Beijing, China, **2** State Key Laboratory of Remote Sensing Science, Beijing Normal University, Beijing, China, **3** Faculty of Geographical Science, Beijing Normal University, Beijing, China

* gad@bnu.edu.cn

**Data Availability Statement:** All relevant data are within the manuscript and its Supporting Information files.

**Funding:** This study was supported by the National Key Research and Development Program of China

## Abstract

Water conservation is an important service function of ecosystems. A timely understanding of dynamic changes in the water conservation function is important for the protection and reconstruction of water resources. Based on remote sensing data, meteorological data, land cover data, and the "Technical Criterion for Ecosystem Status Evaluation" issued by the Ministry of Environmental Protection of the People's Republic of China, a comprehensive evaluation system was designed to assess the water conservation function of the Xiongan New Area from 2005 to 2015. The system created from four aspects, including ecological structure, ecological stress, water balance and landscape ecology. The results showed that from 2005 to 2015, the water conservation function of the Xiongan New Area first decreased and then increased, and the overall trend was upward. The increasing areas were mainly concentrated around Baiyangdian and near the grassland. Among all evaluated indicators, the precipitation compliance rate index fluctuated the most from -16.62 in 2010 to 6.70 in 2015. The evapotranspiration index was the largest in 2010 (6.47) and the smallest in 2005 (3.52). The Temperature Vegetation Dryness Index (TVDI) showed that the drought was the severest in 2010 and the least severe in 2015. However, the other indicators remain relatively stable. From the perspective of the spatial distribution, the water conservation function of the Xiongan New Area was gradually enhanced from north to south.

## Introduction

Water resources are the basis for human survival and development [1]. Water conservation is an important service function of ecosystems, and is defined as the process and ability of the ecosystem to keep moisture in the system under certain temporal and spatial ranges and conditions [2–7]. China has the largest population in the world, while China's per capita water resources are approximately 25% of the global average, and the country has been facing water shortages for a long time [8, 9]. Therefore, strategies for effectively revealing, regulating and utilizing the water conservation function is an important issue to alleviate the continuous reduction in freshwater resources faced by human beings [10].

(grant numbers 2017YFB0504102 and 2017YFC1502402 to Adu Gong) and the National Natural Science Foundation of China (grant number 41671412 to Adu Gong). The funders had no role in study design, data collection and analysis, decision to publish, or preparation of the manuscript.

**Competing interests:** The authors have declared that no competing interests exist

To data, the water conservation function can be calculated by numerous methods, such as the water balance method [11], the soil water storage capacity method [12], the precipitation storage method [13], the canopy interception residual method [14], and the multi-factor regression method [15]. Wang et al. comprehensively analyzed each evaluation method, and the results showed that each method had certain limitations. For example, the soil water storage capacity method ignores the influence of forest evapotranspiration and the effect of different layers, such as the canopy and litter on water storage. The precipitation storage method ignores the influence of surface runoff, and the calculated result of the canopy interception residual method is often larger than the actual water conservation [13]. Stanford University, the Nature Conservancy and the World Wide Fund for Nature jointly developed the InVEST (Integrated Valuation of Ecosystem Services and Tradeoffs) model to assess ecosystem service functions. The water conservation module is an important part of the InVEST model, which is based on the water balance method. Neson [16], Marquès [17] and Wang et al [18] successfully applied the InVEST model to the Willamette River in the USA, Francoli basin in northeastern Spain and Daling River catchment in China, respectively. The applicability of InVEST model and the local suitability of parameters are the key to the reliability of the model. However, the calculation methods and parameters in the InVEST model are mostly based on American standards. When applied to the regions of China, it is necessary to establish a constantly improving database that conforms to the features of regional ecosystem service functions, and use the measured data to correct the model parameters. In addition, data acquisition is not easy and is sensitive to data changes. Furthermore, the InVEST model takes the land use type as a unit, which does not consider the topography and other factors. In summary, each method has certain advantages and limitations, and different evaluation methods and index systems often cause large differences in the results [13]. Currently, there is no universally accepted unified evaluation system.

To evaluate China's ecological status in a convenient manner, the Ministry of Environmental Protection of the People's Republic of China formally issued and implemented the Technical Criterion for Ecosystem Status Evaluation (HJ 192–2015) on March 13, 2015 [19]. The technical criterion provided comprehensive index methods to evaluate various aspects of the ecological environment, including wind and sand fixation, biodiversity and water conservation [20]. In recent years, Chinese scholars have carried out a series of water conservation function evaluations based on this criterion. The research areas include Fujian Province, the upper Yangtze River, Jinan city and Wuyi Mountain. Different evaluation factors (such as rainfall, land cover type and geomorphic type) were applied to different regions [21, 22]. Each method has its advantages and limitations. In this paper, we choose the comprehensive index method to evaluate water conservation functions. This method more convenient than the InVEST model, which does not need localized calibration of model parameters, and the parameters can be adjusted according to the characteristics of the research area whenever necessary.

On April 1, 2017, the Central Committee and the State Council of the Communist Party of China established a new national-level district, the Xiongan New Area, which was deemed another new national zone of significance and was called a millennium plan. Baiyangdian wetland, which is the largest lake wetland on the North China Plain, is known as the "kidney of North China" and is mostly located within the Xiongan New Area [23]. In recent years, due to the combined influence of climate change and human activities, Baiyangdian wetland has faced multiple crises, such as frequent dry deposition, wetland shrinkage, and a gradual decline in ecological functions [24]. Therefore, a comprehensive evaluation of the water conservation function of the Xiongan New Area has important practical significance for the utilization and protection of water resources. To data, there are few studies on the evaluation of the water conservation function of Xiongan New Area. This study comprehensively considered the

characteristics of the Xiongan New Area on the basis of the Technical Criterion for Ecosystem Status Evaluation and established a new indicator system for evaluating the water conservation function.

## Materials and methods

### Study area

The study was conducted in the Xiongan New Area, which is located at the 38˚10'-40˚00' north latitude and 113˚40'-116˚21' east longitude and includes Xiong, Rongcheng and Anxin Counties and other surrounding areas in Baoding city, Hebei Province, China (as shown in Fig 1). This area has great practical and far-reaching historical significance to focus on the function of Beijing's non-capital, to explore new models to optimize development in population-intensive areas, to adjust and optimize the urban layout and spatial structure of Beijing-Tianjin-Hebei and to foster new mechanisms for innovation-driven development. The Xiongan New Area is located in the mid-latitude zone, which has a warm temperate monsoon continental climate with four distinct seasons, an average annual temperature of 11.7˚C, annual sunshine of 2,685 hours, and an average annual rainfall of 551.5 mm. With a total area of 31867.2 hm$^2$, Baiyangdian is the largest lake in Hebei Province, which has 143 lakes with an average annual storage capacity of 1.32 billion cubic meters. The annual average evaporation was 1369 mm, and the evaporation was much larger than the precipitation [25]. Baiyangdian is located in Anxin, Rongcheng, Xiong, Gaoyang and Renqiu Counties, 85% of the lake area is located in Anxin County.

### Data sets

To calculate the land surface temperature (LST), two level 1T Landsat 5 images taken on 17 April 2005 and 17 May 2010, and one level 1T Landsat 8 image taken on 18 May 2015 in the Xiongan New Area were used to conduct this research. The data were all preprocessed by radiometric calibration and atmospheric correction.

The meteorological data were downloaded from the China Meteorological Data Sharing Service System (http://data.cma.cn/), which mainly include daily sunshine hours, precipitation, water vapor pressure, lowest temperature, highest temperature and wind speed. In addition, to calculate the precipitation compliance rate, the precipitation data for 11 consecutive years from 2005 to 2015 were selected.

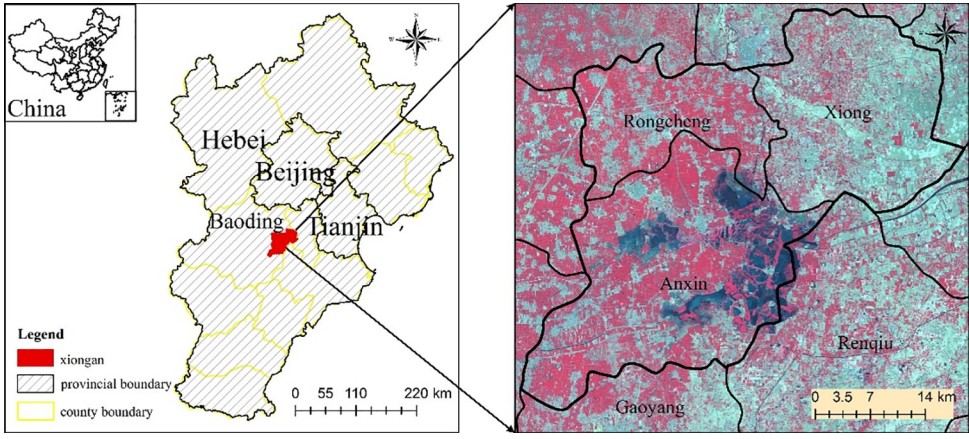

**Fig 1. Study area.**

In this study, a 1:100,000-scale land use/land cover database built by Liu et al. from the Chinese Academy of Sciences was used. The data were based on Landsat images and were interpreted by manual visual observation [26]. The land use/cover data for 2005, 2010 and 2015 were selected. The land cover types were divided into five types: cultivated land, forestland, grassland, wetland and construction land.

## Water conservation function evaluation method

The ecosystem is a complex and expansive system. The various regions are considerably different in terms of natural conditions, economic development status and ecosystem type, making it difficult to determine and quantify the ecological environmental impact factors. Thus, it is difficult to accurately and reasonably evaluate the ecological environmental quality and impact. Currently, there are various regional ecological assessment methods. Among these, the comprehensive index method is the most commonly used. This method can effectively reflect the contribution of each factor to the overall ecological quality status and take into account the adoption and integration of expert opinions when evaluating some non-quantitative factors to reflect the comprehensiveness, integrity and hierarchy of the ecological assessment. The basic principle is to select multiple factors that affect the evaluation object, assign corresponding weights and comprehensively evaluate the evaluated objects. The calculation formula is as follows:

$$R = \sum_{i=1}^{n} R_i x_i \tag{1}$$

where $R$ is the comprehensive index for the evaluation of the water source conservation function, $R_i$ is a quantitative expression of the index factor of the $ith$ water source conservation function evaluation, $x_i$ represents its weight, and n is the number of evaluated indicators.

## Weight determination method

The index weight is a comprehensive measurement of the subjective and objective indication of the relative importance of the indicators in decision-making or assessing problems. Currently, there are many methods for determining weights, which can be divided into subjective and objective weighting methods. The former is based on the evaluator's emphasis on each indicator according to their experience. The latter determines the weight of the indicator based on the amount of inherent information objectively provided by each indicator. Typically, the analytic hierarchy process (Delphi method) [27], principal component analysis [28] and the entropy weight method [29] are commonly used methods.

In this paper, the rank correlation method was selected [30]. First, according to the actual situation, experts can reasonably determine the order of importance of each indicator. Then, the weights are scientifically re-distributed under the premise that the order of the indicators remains unchanged in terms of importance. This method does not require the construction of a judgement matrix, and the calculation is convenient and practical. The basic steps are detailed in the cited reference [30].

## Evolution analysis method

Evolution analysis is a quantitative analysis of the characteristics and processes of water conservation function during different periods. Time series analysis has been widely used in ecological environments to examine land resource and vegetation changes, and the index-based comparison method is one of the main methods that primarily reflects subtle changes in the evaluation unit index [31].

The index-based comparison method is used to calculate the difference between the two times. A positive value indicates that the water conservation function has increased, a negative value indicates that it has decreased, and 0 shows that the regional water conservation function has not changed.

$$\Delta R = R_b - R_a \tag{2}$$

where $\Delta R$ is the change in the water conservation function during the study period, and $R_a$ and $R_b$ represent the initial and final water conservation function indices, respectively.

## Measurement method for land use change

The rate of land use change can be quantitatively described by the land use dynamics model, which measures the quantity change of a specific land use type in a certain period of time and was calculated as Eq (3) [32].

$$S = \left\{ \sum_{ij}^{n} \left( \frac{\Delta S_{i-j}}{S_i} \right) \times \left( \frac{1}{T} \right) \times 100\% \right\} \tag{3}$$

where $S_i$ is the total area of the i land use type at the start of monitoring, $\Delta S_{i-j}$ is the sum of the area of the i land use type converted to other land-use types from the beginning to the end of monitoring, and $T$ is the time period.

## Establishment of evaluation index system

In 2015, the Ministry of Environmental Protection of the People's Republic of China formally issued the Technical Criterion for Ecosystem Status Evaluation, in which the comprehensive functional index (Functional Ecological Index, FEI) is used to evaluate the ecological function of ecological functional zones. A three-level indicator system was used to evaluate the water conservation function, including two indicators, five sub-indices and 12 sub-indicators (as shown in Table 1).

### Ecological structure index and ecological stress index

The Technical Criterion for Ecosystem Status Evaluation characterizes the water conservation function from both ecological and environmental perspectives. As environmental conditions are difficult to determine and ecological status is the key to water conservation function

**Table 1. Index systems for evaluating water conservation functions (before improvement).**

| Indicator type | Sub-indices | Sub-indicators | Weights | Type |
|---|---|---|---|---|
| Ecological status indicator(0.60) | Ecological function index | Water conservation index | 0.25 | positive |
| | | Proportion of protected area | 0.20 | positive |
| | Ecological structure index | Forestland coverage | 0.15 | positive |
| | | Grassland coverage | 0.10 | positive |
| | | Wetland area ratio | 0.15 | positive |
| | Ecological stress index | Area ratio of cultivated land to construction land | 0.15 | negative |
| Environmental condition indicator (0.40) | Pollution load index | Main pollutant emission intensity | 0.45 | negative |
| | | Pollution source emission compliance rate | 0.10 | positive |
| | | Urban sewage treatment rate | 0.10 | positive |
| | Environment quality index | Water quality compliance rate of Class III and better compared with class III | 0.20 | positive |
| | | Air quality compliance rate | 0.10 | positive |
| | | Water quality compliance rate of centralized drinking water sources | 0.05 | positive |

assessment, this study analyzed the water conservation function only from the ecological perspective. Based on the actual situation of the Xiongan New Area, four indicators were reserved from the technical criterion, including the forestland coverage, grassland coverage, wetland area ratio, and area ratio of cultivated land to construction land, whose calculation formulas are shown in Table 2.

## Water balance indices

The water balance theory is the basis for the analysis of hydrological processes, as well as the guide for the quantity and quality evaluation of water resources. Regional water balance is used to study the movement of water through quantitative analysis of the incoming and water expenditures. In terrestrial ecosystems, atmospheric precipitation is the direct source of regional water conservation. The amount of rainfall directly affects the intensity of the water conservation function, and the rainfall compliance rate was considered to be proportional to the water conservation function within a certain range. In addition, evapotranspiration and drought were considered to be two negative indicators affecting water balance. Moreover, as the Xiongan New Area is located on the North China Plain, which has an elevation of only 7–19 m, runoff is negligible. In summary, three indicators were added, including the precipitation compliance rate, drought index and evapotranspiration index.

**(1) Precipitation compliance rate.** In this study, the precipitation anomaly index was used to reflect the precipitation compliance rate, and the degree of deviation in precipitation during a certain period of time and the average state during the same period were evaluated.

$$\text{Pa}(\%) = \frac{P - \bar{P}}{\bar{P}} \times 100 \tag{4}$$

$$\bar{P} = \frac{1}{n}\sum_{i=1}^{n}P_i \tag{5}$$

where Pa is the precipitation compliance rate, $P$ is the precipitation during a certain period of time, $\bar{P}$ is the average climate precipitation during the same period of time, n is the number of years studied, and $i = 1, 2, 3...n$.

**(2) Evapotranspiration index.** Currently, the methods for estimating regional surface evapotranspiration by remote sensing mainly include the empirical model, single-layer and double-layer models based on the energy balance principle, and physical models based on Penman's formula [33]. The P-M (Penman-Monteith) model is an effective method for directly calculating evapotranspiration that comprehensively considers the energy balance of radiation and sensible heat with aerodynamic transfer equations. This model has a solid physical foundation and has become the standard method for calculating the reference crop evapotranspiration, $ET_0$ [34]. It is expressed as follows:

$$\text{ET}_0 = \frac{0.408\Delta(R_n - G) + \gamma\frac{900}{T+273}u_2(e_s - e_a)}{\Delta + \gamma(1 + 0.34u_2)} \tag{6}$$

**Table 2. Calculation formula of ecological structure index and ecological stress index.**

| Indices name | Indicator name | Calculation formula |
|---|---|---|
| Ecological function index | Forestland coverage | $A_{for}$×forestland area/county area; $A_{for} = 104.43$ |
| | Grassland coverage | $A_{gra}$×grassland area/county area; $A_{gra} = 120.58$ |
| | Wetland area ratio | $A_{wet}$×wetland area/county area; $A_{wet} = 321.44$ |
| Ecological stress index | Area ratio of cultivated land to construction land | $A_{gjd}$× (cultivated land area+ construction land area)/ county area; $A_{gjd} = 102.72$ |

where $ET_0$ is the reference evapotranspiration ($mm \cdot d^{-1}$), $\Delta$ is the rate of change of the saturated vapor pressure with temperature ($kPa \cdot {}^\circ C^{-1}$), $R_n$ is the net radiation ($MJ \cdot m^{-2} d^{-1}$), G is the soil heat flux ($MJ \cdot m^{-2} d^{-1}$), $\gamma$ is the dry-wet table constant ($kPa \cdot {}^\circ C^{-1}$), T is the average temperature at 2 m (${}^\circ C$), $u_2$ is the average wind speed at 2 m ($m \cdot s^{-1}$), $e_s$ is the saturated vapor pressure (kpa), and $e_a$ is the actual vapor pressure (kpa).

**(3) Drought index.** To data, remote sensing methods for monitoring agricultural droughts have mainly included the thermal inertia method [35], combined vegetation index and surface temperature method [36] and the microwave remote sensing method [37]. Among these, the temperature vegetation dryness index (TVDI) has been widely used in drought monitoring, this method was proposed based on the spatial variation characteristics of the surface temperature and vegetation index [38, 39]. In this paper, the combination of NDVI and $T_s$ was used to calculate TVDI, which describes the drought situation of the Xiongan New Area. Currently, there are various surface temperature inversion algorithms, such as the atmospheric correction method, the single channel algorithm and the split window algorithm. In this study, the atmospheric correction method was selected to invert the surface temperature.

$$TDVI = (T_s - T_{smin})/(T_{smax} - T_{smin}) \tag{7}$$

$$T_s = K_2 / \ln \left( \frac{K_1}{B(T_s)} + 1 \right) \tag{8}$$

where $T_s$ is the surface temperature ($K$), $B(T_s)$ is the black body thermal radiance, for Landsat-5, $K_1 = \frac{607.76 W}{m^2 \cdot \mu m \cdot sr}$, and $K_2 = 1260.56\ K$, for Landsat-8, $K_1 = 774.89\ W/(m^2 \cdot \mu m \cdot sr)$, and $K_2 = 1321.08 K$. $T_{smax}$ and $T_{smin}$ represent the highest and lowest values of the surface temperature corresponding to an NDVI value, respectively. The value of TVDI ranges from 0 to 1, where the wet side of the TVDI value range is 0, indicating that the soil water content is almost equal to the field water holding capacity, and the TVDI value of the dry side is 1, indicating that the soil water content is close to the wilting point. The fitting equations for $T_{smax}$ and $T_{smin}$ are as follows:

$$T_{smin} = a_1 + b_1 \times NDVI \tag{9}$$

$$T_{smax} = a_2 + b_2 \times NDVI \tag{10}$$

$$NDVI = \frac{\rho_{NIR} - \rho_R}{\rho_{NIR} + \rho_R} \tag{11}$$

where $a_1$ and $b_1$ are the coefficients of the wet-edge fitting equation, $a_2$ and $b_2$ are the coefficients of the dry-edge fitting equation, and $\rho_{NIR}$ and $\rho_R$ are the near-infrared and red-light reflectance, respectively.

## Landscape ecological indices

Landscape ecology mainly studies the interaction and dynamic changes of ecological processes and spatial patterns of landscapes on a macro scale [40, 41]. The research objects in this field includes the entire landscape and emphasis is placed on the interaction between different ecosystems, the maintenance and development of spatial heterogeneity, the protection and management of environmental resources, and the interference of human activities. The landscape index is the most common method for quantitatively assessing landscape ecology [42]. To data, scholars have proposed a variety of landscape indices, such as landscape fragmentation,

dominance, abundance, agglomeration and Shannon diversity [43]. Combining the characteristics of the study area and the selection principles of the index, the human impact index and landscape fractal dimension were selected to analyze the dynamic changes in the landscape pattern.

**(1) Human impact index.** Disturbances caused by human activities greatly influence changes in landscape patterns and continuously reduce the original natural characteristics of landscape types. According to the type of landscape and the changing characteristics, the human impact index describes the intensity of human disturbances to various landscapes in a certain area. The calculation formula is as follows [44]:

$$HII = A_i P_i / TA \tag{12}$$

where HII is the human influence index, $A_i$ is the area of the i-th landscape component, and $P_i$ is the artificial influence intensity coefficient reflected by the $i$-th landscape component, which can be reflected by the Lohani list method, the Leopold matrix method [45], or the Delphi method [46]. In this study, the average of the three methods was used to reduce error (Table 3). TA is the total area of the landscape. The HII ranges from 0 to 1. The larger the value is, the greater the impact of human activities is on the landscape components, and the lower the natural components of the landscape and vice versa [47].

**(2) Landscape fractal dimension.** The landscape fractal dimension quantitatively describes the size of the core area of a certain landscape type and the tortuosity of its boundary line, revealing the complexity of the landscape pattern and patch on a certain observational scale and indirectly reflecting the degree of disturbance of human activity [48]. In this study, the patch characteristics were used to represent the spatial pattern of the landscape. Therefore, the area-circumference method was used to determine the fractal dimension. The calculation formula is as follows:

$$D = 2 \, ln(P/4)/ln \, (A) \tag{13}$$

where $D$ is the fractal dimension of the landscape type, P is the perimeter of the patch, A is the patch area, and the range of $D$ is [1, 2]. A larger D value indicates that the patch shape is more complex. The closer the D value is to 1, the simpler the patch shape is, indicating a greater degree of interference because of human activity.

## Comprehensive evaluation index of the water conservation function

In summary, four sub-indices, including the ecological structure index, water balance index, ecological stress index and landscape ecological index, and nine sub-indicators were determined to evaluate the water conservation function. Each indicator was weighted according to the rank correlation method, and the indicator type was determined according to the relationship of each index and the water conservation function. The results are shown in

**Table 3. Strength indices of human impact for different landscape elements.**

| Landscape component | Cultivated land | Forestland | Grassland | Wetland | Construction land |
|---|---|---|---|---|---|
| Lohani list method | 0.55 | 0.12 | 0.10 | 0.12 | 0.96 |
| Leopold matrix method | 0.57 | 0.14 | 0.09 | 0.13 | 0.94 |
| Delphi method | 0.65 | 0.13 | 0.11 | 0.16 | 0.91 |
| Average value | 0.59 | 0.13 | 0.10 | 0.14 | 0.94 |

**Table 4. Index systems for evaluating water conservation function (improved).**

| Indicator type | Sub-indices | Sub-indicators | Weights | Type |
|---|---|---|---|---|
| Ecological status indicator | Ecological structure index | Forestland coverage | 0.01 | Positive |
| | | Grassland coverage | 0.01 | Positive |
| | | Wetland area ratio | 0.23 | Positive |
| | Water balance index | Precipitation compliance rate | 0.23 | Positive |
| | | Drought index | 0.18 | Negative |
| | | Evapotranspiration index | 0.18 | Negative |
| | Ecological stress index | Area ratio of cultivated land to construction land | 0.10 | Negative |
| | Landscape ecological index | Human impact index | 0.03 | Negative |
| | | Landscape fractal dimension | 0.03 | Positive |

Table 4.

$$FEI = 0.01 \times \text{forestland coverage} + 0.01 \times \text{grassland coverage} + 0.23 \times \text{wetland area ratio} + 0.23 \times \text{precipitation compliance rate} - 0.18 \times \text{drought index} - 0.18 \times \text{evapotranspiration index} - 0.10 \times \text{area ratio of cultivated land to construction land} - 0.03 \times \text{human impact index} + 0.03 \times \text{landscape fractal dimension} \quad (14)$$

## Results and analysis

### Dynamic change characteristics of land use in the Xiongan New Area from 2005 to 2015

As shown in Fig 2, the main types of land cover in the study area included cultivated land, forestland, grassland, wetland and construction land. The changes in the land cover types from 2005 to 2015 are shown in Table 5. Cultivated land was the most important and main land use type in the Xiongan New Area. From 2005 to 2015, the area of cultivated land exhibited a trend of slow growth and then a rapid decline. The overall dynamic change was -1.11. From 2005 to 2010, the increase in cultivated land was mainly because of reclaimed beaches, and the

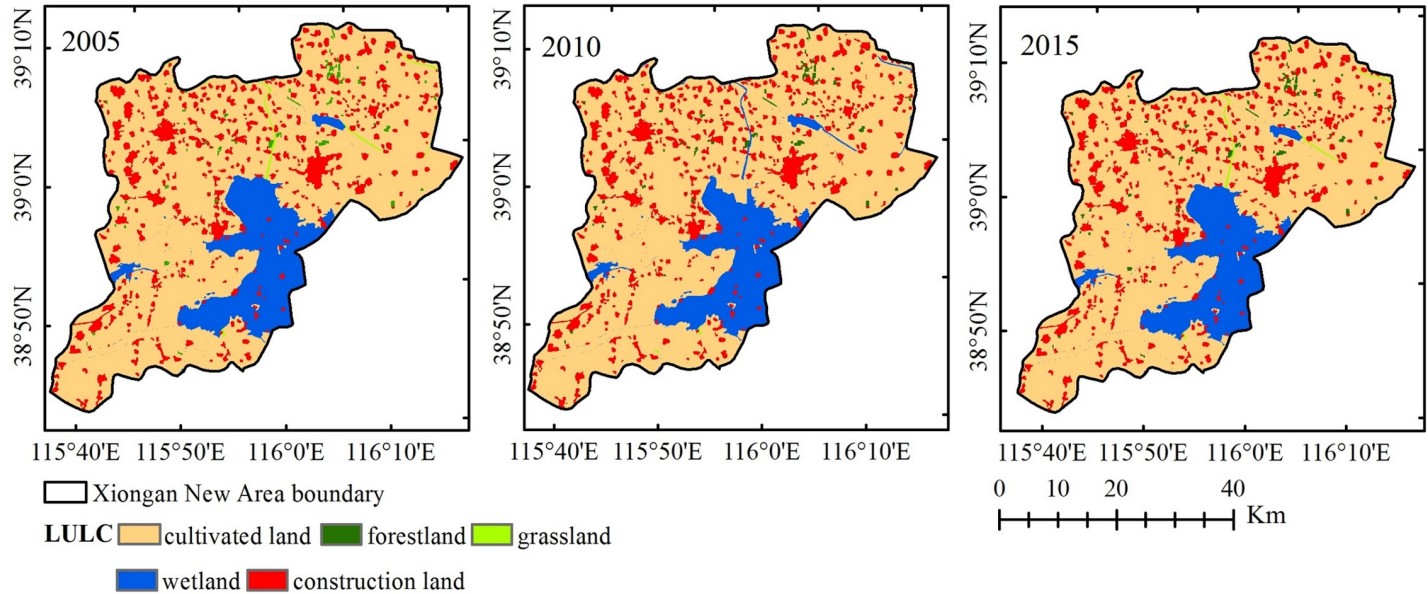

**Fig 2. Land cover map for 2005, 2010 and 2015 in the Xiongan New Area.**

**Table 5. Land cover area and degree of change of the Xiongan New Area in 2005, 2010 and 2015.**

| Land cover | 2005 area (km$^2$) | 2010 area (km$^2$) | 2015 area (km$^2$) | 2005–2010 degree of change | 2010–2015 degree of change | 2005–2015 degree of change |
|---|---|---|---|---|---|---|
| Cultivated land | 1161.61 | 1171.84 | 1032.48 | 0.18 | -2.38 | -1.11 |
| Forestland | 11.34 | 10.98 | 10.17 | -0.63 | -1.48 | -1.03 |
| Grassland | 0.52 | 0.52 | 0.86 | 0 | 13.08 | 6.54 |
| Wetland | 215.52 | 202.12 | 228.43 | -1.23 | 2.60 | 0.60 |
| Construction land | 194.13 | 197.66 | 311.39 | 0.36 | 11.51 | 6.04 |

wetland was converted into cultivated land. However, from 2010 to 2015, the cultivated land area was mainly occupied by expanding construction land, and the farmland was reclaimed into the lake ecosystem. The areas of forestland and grassland were relatively small, and the overall forestland decreased from 11.34 km$^2$ in 2005 to 10.17 km$^2$ in 2015, with a total reduction of 1.17 km$^2$ and a dynamic change of -1.03. The grassland area was maintained at 0.52 km$^2$ from 2005 to 2010, while it increased rapidly to 0.86 km$^2$ by 2015. The overall change during the 10-year period was 6.54. In addition, wetland and construction land were the two other types of land cover except cultivated land. The area of construction land gradually increased. The net growth during the 10 years was 117.26 km$^2$, which was mainly converted from cultivated land area, indicating that the urbanization process was increasing rapidly and that the range of human activities had gradually expanded. The wetland area showed a trend of first decreasing and then increasing, reaching a minimum in 2010 (202.12 km$^2$), which was mainly due to precipitation and beach reclamation. The results were consistent with the research of Zhu et al., who studied the variation characteristics of the Baiyangdian wetland from 1975 to 2018, whose results showed that the wetland area was basically stable from 1975 to 1990, decreased continuously from 1990 to 2011, and increased gradually from 2011 to 2018 [49].

## Trends in the ecological structure index and ecological stress index

According to the above statistical results and meteorological data, four indicators were calculated, including the forestland coverage, grassland coverage, wetland area ratio, and area ratio of cultivated land to construction land. The calculation results are shown in Table 6.

As shown in Table 6, the value of the area ratio of the cultivated land to construction land remained stable, and the normalized results were 0.333, 0.337 and 0.330, respectively, indicating that ecological stress did not change from 2005 to 2015. The reason was that the increase in construction land mainly came from cultivated land, thus the sum of the two remained relatively stable. The coverage rates of forestland were 0.77, 0.75 and 0.69, respectively, with a declining trend. However, the coverage rate of grassland remained stable from 2005 to 2010 and then sharply increased, which may be because the planning and construction of grassland accelerated with the rapid development of urban construction. The wetland area ratios changed relatively little, with values of 45.04, 42.24 and 48.18, respectively.

**Table 6. Ecological structure index and ecological stress index results.**

| Evaluation index / Calculation results | 2005 | | 2010 | | 2015 | |
|---|---|---|---|---|---|---|
| | Result | Normalized | Result | Normalized | Result | Normalized |
| Forestland coverage | 0.77 | 0.348 | 0.75 | 0.339 | 0.69 | 0.312 |
| Grassland coverage | 0.04 | 0.272 | 0.04 | 0.272 | 0.067 | 0.456 |
| Wetland area ratio | 45.04 | 0.332 | 42.24 | 0.312 | 48.18 | 0.356 |
| Area ratio of cultivated land to construction land | 87.97 | 0.333 | 88.86 | 0.337 | 87.20 | 0.330 |

## Trend of the landscape ecological index

Fragstats software was used to calculate the landscape index with the land cover data, and the human impact index and the landscape fractal dimension for each landscape type of the Xiongan New Area in 2005, 2010 and 2015 were obtained.

As shown in Table 7, the cultivated land had the largest human impact index, which was affected by the topography, water source, soil and human cultivation. The cultivated land in the Xiongan New Area was mostly present continuously and had the largest patch area, while the number of patches was the lowest except for those of the grassland. The human impact of construction land gradually increased from 2005 to 2015 with the rapid development of urbanization, and the population moved closer to towns. As Baiyangdian is the main wetland in the Xiongan New Area and the area changes mainly due to cultivation and precipitation, the wetlands were less affected by human activities. Moreover, the forestland and grassland areas were small; thus, human activities had the least impact on them. In summary, the overall impact of human activities increased from 0.586 in 2005 to 0.607 in 2015.

According to the 2005–2015 landscape fractal dimension index, the values of cultivated land and wetland increased from 1.319 to 1.359 and 1.276 to 1.507, respectively, indicating that the patch patterns of cultivated land and wetland were becoming increasingly complicated. In particular, the wetland landscape area exhibited a large increase in 2015 that was closely related to precipitation. The fractal dimension of construction land decreased from 1.313 to 1.256, indicating that humans have consciously planned the expansion of the construction land during the process of urbanization. The value of forestland first decreased and then increased, which was related to land degradation and the gradual reduction in forestland area. In addition, the grassland area was small; thus, the landscape dimension was neglected.

## Variation characteristics of the water balance index

**Trends in the evapotranspiration index.**   The land surface temperature (LST) and NDVI were calculated as Eq (8) and Eq (11), then the dry-wet edge equation was fitted. According to the fitted dry-wet edge equation, the TDVI values of 2005, 2010 and 2015 were calculated. The drought level was divided into 5 levels based on the drought situation of the study area and national meteorological drought grade standards, namely, wet (0–0.2), normal (0.2–0.4), light drought (0.4–0.6), moderate drought (0.6–0.8) and heavy drought (0.8–1.0). The distributions of drought in the Xiongan New Area in 2005, 2010 and 2015 are shown in Fig 3.

As shown in Fig 3, from the perspective of the time series changes, the TVDI changed considerably from 2005 to 2015. Fig 4 shows that the fitting accuracies of the dry and wet edges in 2005 and 2015 were higher, the accuracies of the dry edge reached 0.940 and 0.774,

**Table 7. Landscape index results.**

| Landscape index | Year | Cultivated land | Forestland | Grassland | Wetland | Construction land |
|---|---|---|---|---|---|---|
| Number of patches | 2005 | 38 | 54 | 2 | 437 | 381 |
|  | 2010 | 48 | 57 | 2 | 471 | 404 |
|  | 2015 | 35 | 51 | 3 | 439 | 442 |
| Human impact index | 2005 | 0.446 | $9.584 \times 10^{-4}$ | $3.381 \times 10^{-5}$ | 0.020 | 0.119 |
|  | 2010 | 0.450 | $9.280 \times 10^{-4}$ | $3.381 \times 10^{-5}$ | 0.018 | 0.121 |
|  | 2015 | 0.396 | $8.596 \times 10^{-4}$ | $5.591 \times 10^{-5}$ | 0.021 | 0.190 |
| Landscape fractal dimension | 2005 | 1.319 | 1.315 | / | 1.276 | 1.313 |
|  | 2010 | 1.322 | 1.179 | / | 1.293 | 1.264 |
|  | 2015 | 1.359 | 1.634 | / | 1.507 | 1.256 |

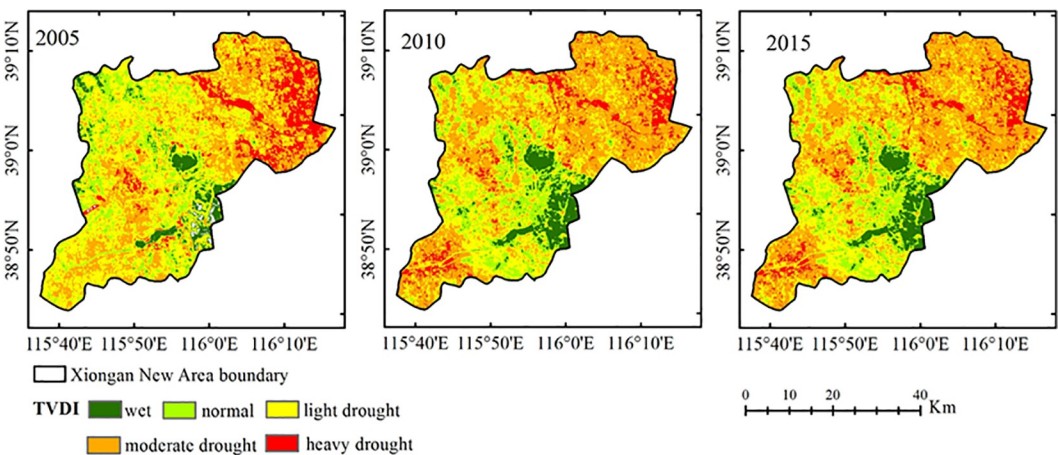

**Fig 3. Distribution of TVDI in the Xiongan New Area in 2005, 2010 and 2015.**

respectively, and the wet edge were 0.711 and 0.849. The dry edge accuracy was higher at 0.768 in 2010, while the wet edge accuracy was only 0.196. Previous studies showed that the LST and NDVI on the dry edge exhibited a significant negative correlation, indicating that the LST decreased with the increase in vegetation coverage when the vegetation was under water stress. However, most of the LST and NDVI on the wet side showed a positive or irrelevant relationship, which was consistent with the principle of TVDI, indicating the validity of the results of this study. Fig 5 shows that the overall trend of the 3-year LST spatial distribution was approximately similar, with higher temperatures in the densely populated areas with buildings and lower temperatures in the vicinity of water. Since the temperature in May was higher than that in April, the temperature ranges were 284–310 K, 292–315 K and 295–318 K, respectively. According to the NDVI distribution (Fig 6), the NDVI values were positive for the water areas and higher in 2005 than in 2010 and 2015. This finding was mainly because the water was full of lotus leaves, and May had more vegetation than April. As shown in Fig 3, the normal area was the largest in 2005, followed by the light drought area, accounting for 37.62% and 33.08% of the total area, respectively. The moderate drought area was the largest in 2010, reaching 697.94 $km^2$ and accounting for 45.83% of the total area, and the normal area ratio was reduced

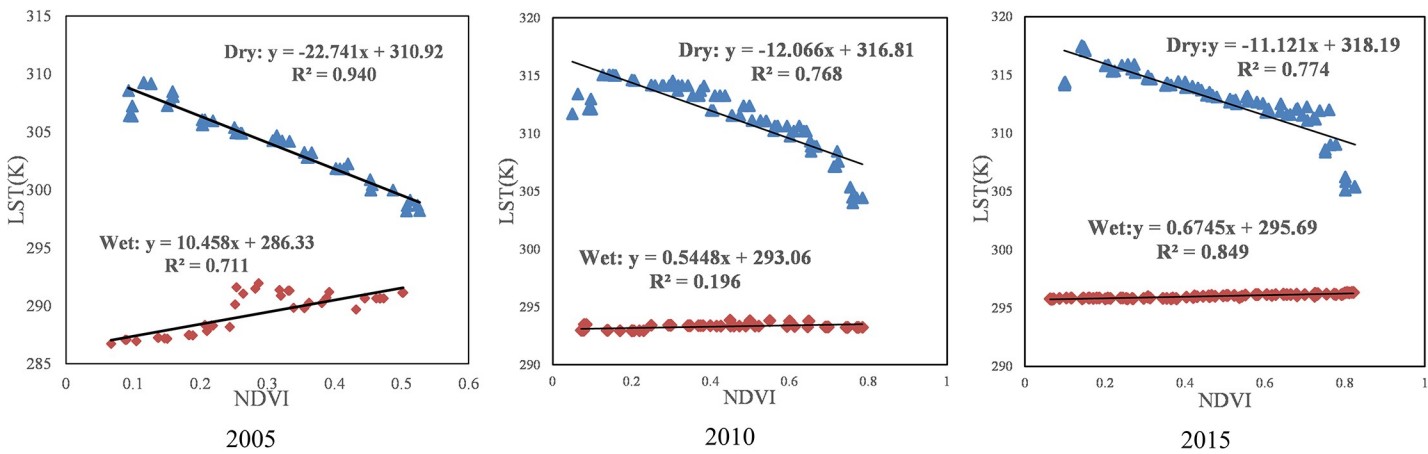

**Fig 4. Fitting results for LST and NDVI for the Xiongan New Area in 2005, 2010 and 2015.**

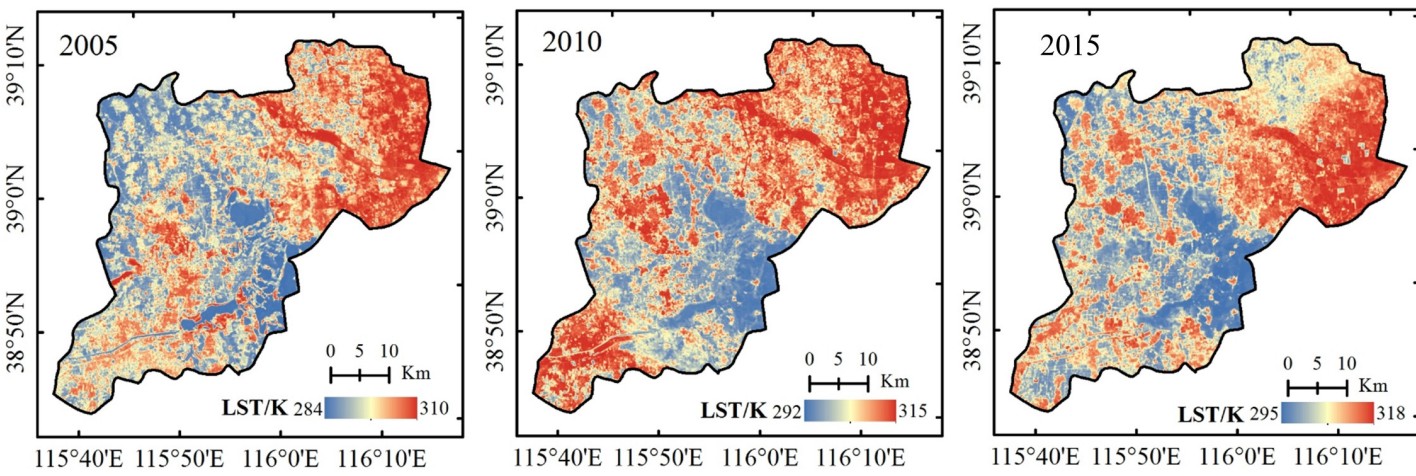

**Fig 5. Distribution of land surface temperature in the Xiongan New Area in 2005, 2010 and 2015.**

to 26.37%. The drought improved in 2015, the wet area increased, and the normal area was the largest, accounting for 36.61% of the total area. In summary, drought was the most serious in 2010, the least serious in 2015 and moderate in 2005.

**Trend of precipitation compliance rate and evapotranspiration index.** Based on the meteorological data, the evapotranspiration index and precipitation compliance rate were calculated.

As shown in Table 8, the precipitation compliance rates were -2.02, -16.62 and 6.70, respectively, and were the most volatile of all indicators. The annual average precipitation from 2005 to 2015 was 500.0 mm, and the annual rainfall for 2005, 2010 and 2015 was 489.9 mm, 416.9 mm and 533.5 mm, respectively, indicating that 2005 and 2010 were dry years, especially 2010, and 2015 was a wet year.

The evapotranspiration calculation results showed that evapotranspiration was the highest in 2010, and the lowest in 2005. The evapotranspiration was calculated based on the FAO56 P-M model, which was achieved via plant and soil transport processes and was mainly related to meteorological factors, such as wind speed and temperature. This result was mainly because

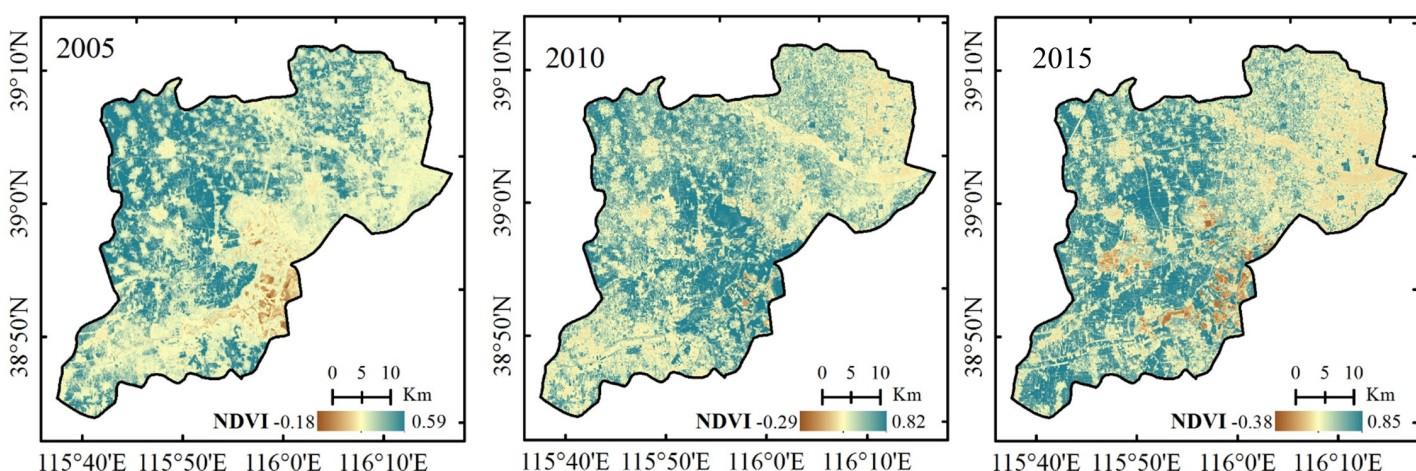

**Fig 6. Distribution of NDVI in the Xiongan New Area in 2005, 2010 and 2015.**

**Table 8. The results of evapotranspiration index and precipitation compliance rate.**

| Calculation results<br>Evaluation indices | 2005 | | 2010 | | 2015 | |
|---|---|---|---|---|---|---|
| | Result | Normalized | Result | Normalized | Result | Normalized |
| Evapotranspiration index | 3.52 | 0.234 | 6.47 | 0.430 | 5.07 | 0.336 |
| Precipitation compliance rate | -2.02 | -0.080 | -16.62 | -0.656 | 6.70 | 0.264 |

the average temperature, wind speed and sunshine duration in 2005 were relatively low, but the humidity was relatively high.

## Changing trend of the water conservation function in the Xiongan New Area

Based on the calculated result of each indicator and Eq (14), the water conservation functions for the Xiongan New Area in 2005, 2010 and 2015 were calculated, and then the changing trends from 2005 to 2010, 2010–2015 and 2005–2015 were calculated, as shown in Figs 7 and 8, respectively.

As shown in Fig 7, the FEI values for the water conservation function were the highest in 2015, moderate in 2005, and the lowest in 2010. In 2005and 2010, the FEI values were all negative, and the average values were -0.114 and -0.289, respectively, indicating that the water conservation function was low and that the ecological environment was poor. By 2015, the FEI ranged from -0.12 to 0.07, the average value was -0.02, which was significantly higher than that in 2010. From the perspective of the spatial distribution, the areas with poor water conservation function were mainly distributed throughout Xiong County and the urban centers in Rongcheng and Anxin Counties. The areas around Baiyangdian had better water conservation functions.

Among all the indicators that showed a positive correlation, the grassland coverage, wetland area ratio and precipitation compliance rate were the highest in 2015; in particular, the precipitation compliance rate was positive in 2015 and negative in 2005 and 2010, exhibiting the most change among all years. Moreover, the wetland area ratio and precipitation compliance rate were the two indices with the highest weight among all indicators, both of which were 0.23. The forestland coverage was the smallest in 2015; however, the difference was small compared with those in 2005 and 2010. Additionally, the amount of forestland coverage was small; thus, the impact was small. The drought index, evapotranspiration index, area ratio of cultivated land to construction land and human impact index were four negatively correlated indicators. Among them, the evapotranspiration index and drought index were the most negative indicators affecting the water conservation function, with weight coefficients of -0.18. The evapotranspiration index was the smallest in 2005, the largest in 2010, and moderate in 2015. The

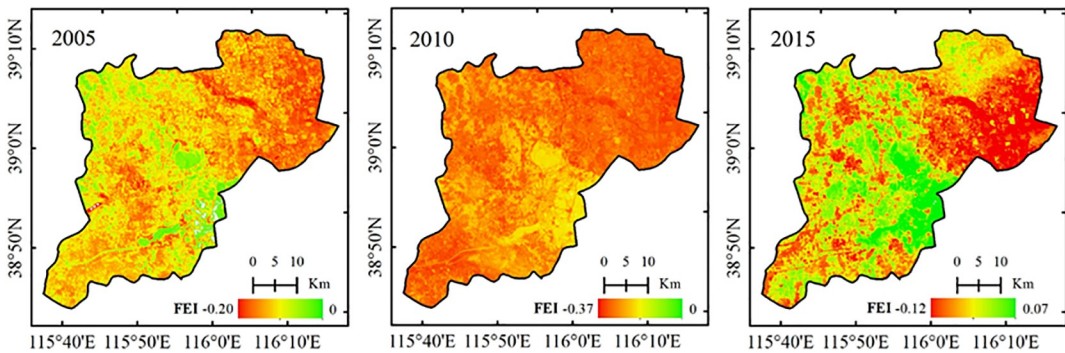

**Fig 7. Water conservation function for the Xiongan New Area in 2005, 2010 and 2015.**

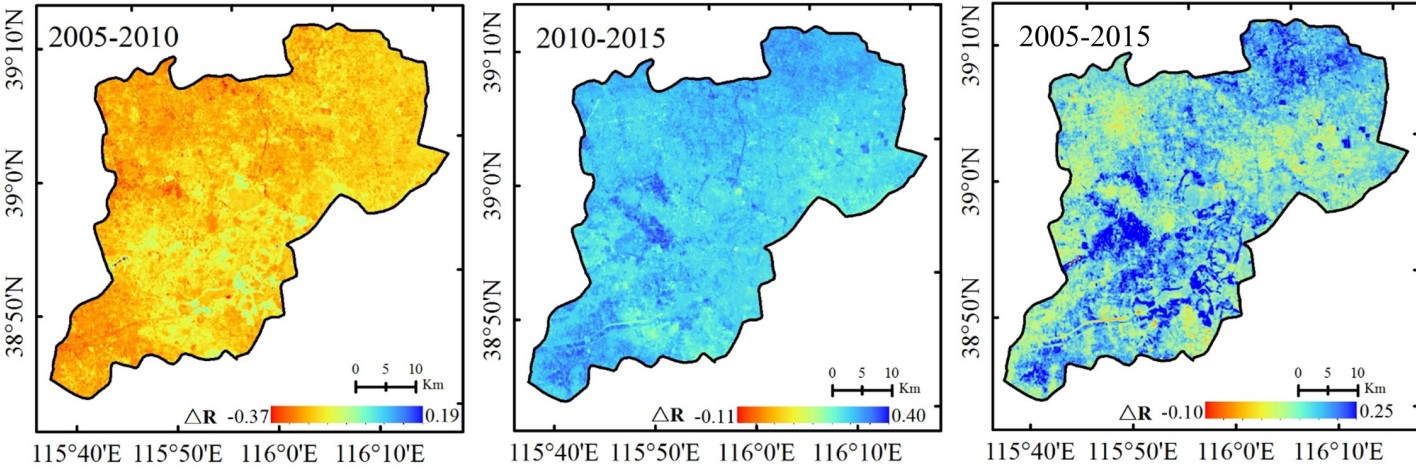

**Fig 8. Change in the water conservation function for the Xiongan New Area from 2005 to 2010, 2010 to 2015 and 2010 to 2015.**

drought index was the smallest in 2015 and the largest in 2010. The weight coefficient of the area ratio of cultivated land to construction land was -0.10, which was the third largest impact indicator and remained relatively stable. The human impact index showed an upward trend from 2005 to 2015. In addition, from the perspective of the spatial distribution, the water conservation function of the Xiongan New Area was gradually enhanced from north to south, which was the strongest near the wetland, and the construction land was the smallest.

Fig 8 shows that the ΔR values were all negative from 2005–2010, indicating that the water conservation function was significantly worse. However, the ΔR values were positive from 2010 to 2015, indicating a significant improvement in the ecological environment. From 2005–2015, the ΔR values displayed first decreased and then increased, and the overall trend was increasing, indicating that the ecological environment had improved. This result was basically consistent with the results of Yang et al., who evaluated the ecological quality of the Xiongan New Area from 1995 to 2015 [50].

## Discussion

The water conservation function around Baiyangdian showed a trend of decline and then increase, which was mainly related to climate and policy. Baiyangdian is a semi-enclosed lake on the plain that cannot self-regulate. The water quantity of Baiyangdian is mainly derived from precipitation and the inflow of upstream rivers. There was a positive correlation between wetland area and precipitation. In 2005 and 2010, the rainfall was significantly lower than the 10-year average rainfall, while it was higher than the average in 2015. Since the beginning of the 1960s, to remediate the Haihe River, more than 100 reservoirs have been built in the upper reaches of the Baiyangdian. The considerable construction of reservoirs has reduced the runoff of the upstream river channel, changing the spatial and temporal distribution of water resources, and decreasing the amount of water entering the lake. As a result, the water supply to the Baiyangdian became insufficient, resulting in the continuous reduction of the Baiyangdian area and the repeated occurrence of dry deposition [51]. From 1981 to 2007, the Hebei provincial government successively replenished Baiyangdian with water from upstream 21 times. The total water replenishment was approximately $6.36 \times 10^8$ $m^3$, which alleviated the degradation of the Baiyangdian wetland to some extent. However, the Baiyangdian wetland still showed a continuous declining trend. In December of 2005, the Hebei provincial government

launched a 10-year ecological environment management plan to revitalize Baiyangdian. This plan, together with the South-to-North Water Diversion Project, continued to return water to Baiyangdian, resulting in an increase in the lake area in 2015.

In recent years, the upstream area and surrounding economy of Baiyangdian has developed rapidly and the population has increased sharply. By 2013, the resident population of Baoding city reached 11.07 million, an increase of 10.51 million compared with the population of 0.90 million in 1984. In addition, the urbanization of the Xiongan New Area has been rapid, and the construction land area increased from 194.13 $km^2$ in 2005 to 311.39 $km^2$ in 2015. The above metrics show that human activities and ecological stress have continued to enhance, resulting in gradually worsening water conservation function around the town.

Under the circumstance of a declining ecological environment, the grassland in the Xiongan New Area have been highly valued and strictly protected. The local government has carried out large-scale landscaping to improve the ecological quality of grassland. As a result, the water conservation function around grassland was gradually enhanced.

Compared with the commonly used evaluation methods, the research method of this paper had its own advantages. The InVEST model is currently the most widely used ecosystem evaluation model, but the calculation methods and parameters in the model were mostly based on American standards. When applied to China, it is necessary to establish a constantly improving database that conforms to the features of regional ecosystem service functions, and use the measured data to correct the model parameters. The Remote Sensing Ecological Index (RESI) index proposed by Xu is a simple ecological quality assessment method that has been widely used in many research areas [52], but the results show that the index is more suitable for terrestrial regions. In this study, Baiyangdian occupied a large area of the Xiongan New Area, and this method was therefore not suitable. In the comprehensive index method used in this study, the selection of impact factors is based on the characteristics of the study area, the research results are closer to reality, and the operability and generalization of the method are stronger. However, this study also had certain limitations. First, the weight coefficient was determined by the rank correlation method, which is subjective. In addition, the three assessed months are different, which had a certain influence on the research results.

## Conclusion

A timely understanding of the dynamic changes in the water conservation function was conducive to the protection and reconstruction of water resources. Based on the Technical Criterion for Ecosystem Status Evaluation and characteristics of the study area, a new comprehensive evaluation system was designed to assess the water conservation function of the Xiongan New Area from 2005 to 2015. The results show that the water conservation function of the Xiongan New Area first decreased and then increased, and the overall trend was upward, indicating that the ecological environment had improved. The increasing areas were mainly concentrated around Baiyangdian and near the grassland, while the surrounding towns gradually decreased with the development of urbanization. Among the indices, the ecological structure and ecological stress indices performed stably, while the water balance and landscape ecological indices had relatively lager fluctuations, indicating that the impacts of the climate and human activities were large.

## Supporting information

**S1 File.**
(RAR)

## Acknowledgments

Foremost, we would like to thank Dr. Zheng Wang from Peking University and Dr. Hongkui Zhou from Beijing Normal University who made some comments on this article.

## Author Contributions

**Conceptualization:** Yanling Chen.

**Formal analysis:** Yanling Chen.

**Funding acquisition:** Adu Gong.

**Investigation:** Yanling Chen, Adu Gong, Tingting Zeng, Yuqing Yang.

**Methodology:** Yanling Chen, Adu Gong.

**Validation:** Yanling Chen.

**Writing – original draft:** Yanling Chen.

**Writing – review & editing:** Adu Gong, Tingting Zeng, Yuqing Yang.

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
