## [Decision Letter · Decision Letter 0]

20 Sep 2019

PONE-D-19-19149

Evaluation of Water Conservation Functions in the Xiongan New Area Based on a Comprehensive Index Method

PLOS ONE

Dear Mrs. Chen,

Thank you for submitting your manuscript to PLOS ONE. After careful consideration, we feel that it has merit but does not fully meet PLOS ONE’s publication criteria as it currently stands. Therefore, we invite you to submit a revised version of the manuscript that addresses the points raised during the review process.

We have received two reviewers for your paper. It is recognized that your work assembles a substantial range of data from different sources, and this opens excellent opportunities to explore more the results and provide a more insightful discussion. Reviewer #1 provided essential suggestions, which should be considered in your study. Although the novelty of your work is not a reason for rejection in PLOS One, the observations of reviewer #2 should be used to improve the methodology, graphics, and clarity of your manuscript. Please, I would like to remind you that you are not obliged to add any references provided by the reviewers. As long as you address the concern on update the literature, it is up to you what references you choose to add.

We would be willing to reconsider your manuscript if you are prepared to undertake a major overhaul, addressing these concerns, and re-submitting it. Please note that re-submitting your manuscript does not guarantee eventual acceptance, and that your re-submission will be subject to a new peer-review process before a decision is made.

We would appreciate receiving your revised manuscript by Nov 04 2019 11:59PM. To enhance the reproducibility of your results, we recommend that if applicable you deposit your laboratory protocols in protocols.io, where a protocol can be assigned its own identifier (DOI) such that it can be cited independently in the future. For instructions see: http://journals.plos.org/plosone/s/submission-guidelines#loc-laboratory-protocols

We look forward to receiving your revised manuscript.

Kind regards,

Rodolfo Nóbrega

Academic Editor

PLOS ONE

Journal Requirements:

"The funders had no role in study design, data collection and analysis, decision to publish, or preparation of the manuscript"

Please provide an amended Funding Statement that declares *all* the funding or sources of support received during this specific study (whether external or internal to your organization) as detailed online in our guide for authors at http://journals.plos.org/plosone/s/submit-nowPlease state what role the funders took in the study.  If any authors received a salary from any of your funders, please state which authors and which funder. If the funders had no role, please state: "The funders had no role in study design, data collection and analysis, decision to publish, or preparation of the manuscript."

3. Please ensure that you refer to Figures 1, 3 and 4 in your text as, if accepted, production will need this reference to link the reader to the figure.

4. We note you have included tables to which you do not refer in the text of your manuscript. Please ensure that you refer to Tables 1, 2 and 8 in your text; if accepted, production will need this reference to link the reader to each Table.

Reviewers' comments:

Reviewer's Responses to Questions

**Comments to the Author**

1. Is the manuscript technically sound, and do the data support the conclusions?

Reviewer #1: Partly

Reviewer #2: Partly

2. Has the statistical analysis been performed appropriately and rigorously? 

Reviewer #1: Yes

Reviewer #2: No

3. Have the authors made all data underlying the findings in their manuscript fully available?

Reviewer #1: Yes

Reviewer #2: Yes

4. Is the manuscript presented in an intelligible fashion and written in standard English?

Reviewer #1: Yes

Reviewer #2: No

5. Review Comments to the Author

Reviewer #1: Manuscript number: PONE-D-19-19149

Title: Evaluation of Water Conservation Functions in the Xiongan New Area Based on a Comprehensive Index Method

Journal name: PLOS ONE

The authors evaluate the water conservation functions in the Xiongan New Area during 2005, 2010 and 2015 in four areas (ecological structure, ecological stress, water balance and landscape ecology) using a comprehensive evaluation system based on on remote sensing data, meteorological data, land cover data, and the Technical Criterion for Ecosystem Status Evaluation. I would like to congratulate the authors for a considerable amount of work that they have done. In general, the manuscript has got some potential. Indeed, the topic discussed in this manuscript is in high interest. I suggest the author may consider the following comments before the manuscript would become a potential publication. Please consider the particular comments listed below!

1. Introduction

There has already been a large amount of literatures discussing this topic, there is a need to better elaborate the contribution of the work to the existing literature. Information of the models existing and the differences among these and the novel model, what are the advantages and disadvantages of the approach, usage, develop and in general why your model is better, lacks of arguments in the introduction section. Please, be more critical in addressing the research gap. What is the contribution of the paper to the literature? Emphasize these aspects already in the literature reviews to make paper attractive for readers. Some references should be included. e.g., (i) Evaluating sustainability of water-energy-food (WEF) nexus using an improved matter-element extension model: A case study of China. Journal of Cleaner Production. 2018;202:1097-106 (ii) Decoupling analysis of economic growth from water use in City: A case study of Beijing, Shanghai, and Guangzhou of China. Sustainable cities and society. 2018;41:86-94. (iii) Evaluating water resource sustainability in Beijing, China: Combining PSR model and matter-element extension method. Journal of Cleaner Production. 2019;206:171-9.

2. Study area and data sets

Few articles have made this part a separate section. Please read the latest published papers in PLOS ONE carefully and re-arrange the section according to the format of the previous paper in PLOS ONE. It is recommended to merge the Study areas and data sets with the Methodology as one section.

3. Methodology

It is recommended to merge the Study areas and data sets with the Methodology as one section.

4. Results and analysis

It is seemed that there is only a results statement, no discussion or analysis. I should suggest to add an discuss what your findings are different from the past works.

5. Conclusion

This section is well written. I would suggest to write more explicitly the main findings of this study.

6. References

Please check the references in the text and the list; References should be revised and upgraded.

7. Grammar check

There are still some occasional grammar errors through the manuscript especially the article ''the'', ''a'' and ''an'' is missing in many places, please make a spellchecking in addition to these minor issues.

Good luck!

Reviewer #2: The article has no innovations in terms of ideas and data processing, the data is not new enough, the pictures in the text are not clear enough, the references are not new, and the guiding significance is not strong.

6. PLOS authors have the option to publish the peer review history of their article (what does this mean?). If published, this will include your full peer review and any attached files.

Reviewer #1: No

Reviewer #2: No

---

## [Author Response · Author response to Decision Letter 0]

25 Nov 2019

Dear Editors and Reviewers:

Thank you for your letter and for the reviewers’ comments concerning our manuscript entitled “Evaluation of Water Conservation Functions in the Xiongan New Area Based on a Comprehensive Index Method” (No.PONE-D-19-19149).Those comments are all valuable and very helpful for revising and improving our paper, as well as the important guiding significance to our researches. We have studied comments carefully and have made correction which we hope meet with approval. Revised portion are marked in red in the paper. The main corrections and responds to the reviewers’ comments are as follows.

Editor

Response: We are sure that our manuscript meets PLOS ONE's style requirements.

"The funders had no role in study design, data collection and analysis, decision to publish, or preparation of the manuscript".

Response: We agree with you.

3. Please ensure that you refer to Figures 1, 3 and 4 in your text as, if accepted, production will need this reference to link the reader to the figure.

Response: We have made correction according to the Editor’s suggestion. We have checked and revised Figures 1, 3 and 4 in our text so that readers can link to the figures.

4. We note you have included tables to which you do not refer in the text of your manuscript. Please ensure that you refer to in your text; if accepted, production will need this reference to link the reader to each Table.

Response: We have made correction according to the Editor’s suggestion. We have checked and revised Tables 1, 2 and 8 in our text so that readers can link to the Tables.

Reviewer #1:

1. Introduction. There has already been a large amount of literatures discussing this topic, there is a need to better elaborate the contribution of the work to the existing literature. Information of the models existing and the differences among these and the novel model, what are the advantages and disadvantages of the approach, usage, develop and in general why your model is better, lacks of arguments in the introduction section. Please, be more critical in addressing the research gap. What is the contribution of the paper to the literature? Emphasize these aspects already in the literature reviews to make paper attractive for readers. Some references should be included. e.g., (i) Evaluating sustainability of water-energy-food (WEF) nexus using an improved matter-element extension model: A case study of China. Journal of Cleaner Production. 2018;202:1097-106 (ii) Decoupling analysis of economic growth from water use in City: A case study of Beijing, Shanghai, and Guangzhou of China. Sustainable cities and society. 2018;41:86-94. (iii) Evaluating water resource sustainability in Beijing, China: Combining PSR model and matter-element extension method. Journal of Cleaner Production. 2019;206:171-9.

Response: We have rewritten this part according to the Reviewer’s comments. We have rewrote the Introduction section by reading a large amount of literature. We have added the significance of studying the water conservation function, the existing research methods, the advantages and disadvantages of each method, the reasons for choosing this research method, and also explained the significance of Xiongan New Area was selected as the study area. In addition, we have added many classic articles on water resources according to Reviewer’s advice.

2. Study area and data sets. Few articles have made this part a separate section. Please read the latest published papers in PLOS ONE carefully and re-arrange the section according to the format of the previous paper in PLOS ONE. It is recommended to merge the Study areas and data sets with the Methodology as one section.

Response: We have made correction according to the Reviewer’s suggestion. I have read the latest published papers in PLOS ONE carefully and merged the Study areas and data sets with the Methodology as one section.

3. Methodology. It is recommended to merge the Study areas and data sets with the Methodology as one section.

Response: We have made correction according to the Reviewer’s suggestion. We merged the Study areas and data sets with the Methodology as one section.

4. Results and analysis. It is seemed that there is only a results statement, no discussion or analysis. I should suggest to add an discuss what your findings are different from the past works.

Response: We have made correction according to the Reviewer’s suggestion. Firstly, we revised the results and analysis sections to make the results clearer and the presentations smoother. In addition, we have added the discussion section, mainly to analyze the reasons for the results, and pointed out the advantages and limitations of the research method compared with other methods. Please refer to the discussion section for details.

5. Conclusion. This section is well written. I would suggest to write more explicitly the main findings of this study.

Response: We have made correction according to the Reviewer’s suggestion. We have revised the conclusions to highlight the findings of this paper.

6. References. Please check the references in the text and the list; References should be revised and upgraded.

Response: We have made correction according to the Reviewer’s suggestion. We have checked every reference carefully, and since some new literatures were added, the references were updated.

7. Grammar check. There are still some occasional grammar errors through the manuscript especially the article ''the'', ''a'' and ''an'' is missing in many places, please make a spellchecking in addition to these minor issues.

Response: We have made correction according to the Reviewer’s suggestion. I have revised the entire manuscript word for word, and I have made a spellchecking. I did find some grammatical errors and made corrections.

Reviewer #2:

The article has no innovations in terms of ideas and data processing, the data is not new enough, the pictures in the text are not clear enough, the references are not new, and the guiding significance is not strong.

Response: We have made correction according to the Reviewer’s suggestion. (1)We agree with the opinion of reviewer. This study cannot be regarded as a method innovation in a strict sense. The method of this paper was improved on the“Technical Criterion for Ecosystem Status Evaluation”issued by Ministry of Environmental Protection of the People’s Republic of China, and added some new indicators, considering the actual situation of the study area and the principle of water conservation. In addition, Xiongan New Area is deemed another new national zone of significance after the Shenzhen Special Economic Zone and the Shanghai Pudong New Area. It is a millennium plan and a national event. Baiyangdian Wetland is the largest lake wetland in the North China Plain, which is known as the “kidney of North China”，and most of which are located in Xiongan New Area. Studying the trend of water conservation function in Xiongan New Area during the past ten years is conducive to supporting the environmental assessment of government and better protecting Baiyangdian wetland. (2) Since the land use/cover data is only updated to 2015, in order to ensure the uniformity of the data, this study does not include other auxiliary data, and new data will be added in the future research.

(3)We have corrected all the pictures in the article to achieve the optimum clarity of the pictures.

(4) We have revised and updated the references and added plenty of the latest literature.

---

## [Decision Letter · Decision Letter 1]

21 Jan 2020

PONE-D-19-19149R1

Evaluation of Water Conservation Functions in the Xiongan New Area Based on a Comprehensive Index Method

PLOS ONE

Dear Mrs. Chen,

Thank you for submitting your manuscript to PLOS ONE. After careful consideration, we feel that it has merit but does not fully meet PLOS ONE’s publication criteria as it currently stands. Therefore, we invite you to submit a revised version of the manuscript that addresses the points raised during the review process.

After analyzing the reviewers' comments, I would like to draw your attention to the fact that they have raised serious methodological concerns, which I agree and that leaves your manuscript on the borderline between major revision and rejection. I suggest undertaking a major overhaul, addressing all reviewers' concerns and substantially upgrade the present manuscript.  Please note that re-submitting your manuscript does not guarantee eventual acceptance and that your re-submission will be subject to further review before a decision is made.

We would appreciate receiving your revised manuscript by Mar 06 2020 11:59PM. To enhance the reproducibility of your results, we recommend that if applicable you deposit your laboratory protocols in protocols.io, where a protocol can be assigned its own identifier (DOI) such that it can be cited independently in the future. For instructions see: http://journals.plos.org/plosone/s/submission-guidelines#loc-laboratory-protocols

We look forward to receiving your revised manuscript.

Kind regards,

Rodolfo Nóbrega

Academic Editor

PLOS ONE

Reviewers' comments:

Reviewer's Responses to Questions

**Comments to the Author**

1. If the authors have adequately addressed your comments raised in a previous round of review and you feel that this manuscript is now acceptable for publication, you may indicate that here to bypass the “Comments to the Author” section, enter your conflict of interest statement in the “Confidential to Editor” section, and submit your "Accept" recommendation.

Reviewer #1: All comments have been addressed

Reviewer #2: (No Response)

Reviewer #3: (No Response)

2. Is the manuscript technically sound, and do the data support the conclusions?

Reviewer #1: Yes

Reviewer #2: Partly

Reviewer #3: Partly

3. Has the statistical analysis been performed appropriately and rigorously? 

Reviewer #1: Yes

Reviewer #2: I Don't Know

Reviewer #3: Yes

4. Have the authors made all data underlying the findings in their manuscript fully available?

Reviewer #1: Yes

Reviewer #2: No

Reviewer #3: No

5. Is the manuscript presented in an intelligible fashion and written in standard English?

Reviewer #1: Yes

Reviewer #2: No

Reviewer #3: No

6. Review Comments to the Author

Reviewer #1: The authors have incorporated comments from the first round of review. My concerns from my previous review have been addressed. I would recommend the paper to be accepted for publication.

Reviewer #2: (No Response)

Reviewer #3: What's the innovation of this paper? the significance? It's still not clear. The authors should summarized and showed it in a certain part in the paper, maybe in the end of the introduction.

1. The language should be more carefully corrected throughout the whole manuscript.

E.g. in the abstract,

Line 16-17, "The system created from four aspects, namely ecological structure, ecological stress, water balance and landscape ecology, and which includes nine indexes " What "and which" means?

Line 19-21 " The increasing areas are mainly concentrated around the Baiyangdian and near the grassland, while the towns surrounding are gradually decreasing along with the development of urbanization." The towns...decreasing? How?

Line 23-26, what's the difference between "the most","the strongest" and "the largest"?

Line 25-26, " The Temperature Vegetation Dryness Index (TVDI) shows that the most severe drought in 2010 and the lightest in 2015."

2. Introduction

The introduction is long but cannot really introduce the reader to read your article. It should be narrated mainly around your aim of this manuscript. Too many literatures were just listed in this part.

Line 31-44, this paragraph may be shortened to just show the significance of this study.

Line 45-63, the use of RS, the authors take so much sentences on the use of RS, Does it the main method or the features of this paper? What's more, so many literatures were just listed here.

And it is the same with nest paragraph.

Another paragraph should be added to show how this paper to tackle this problems.

Data sets

Line 120-121, Why data in April and May were chosen? and What about data in summer or the other season? and Why it wasn't in the same month?

Line 130-131, What's the relation between the data of all the time nodes?

Line 159, What's GI stand for?

Line 193, What's the diference between woodland and forestland? they both appeared in this paper.

Conclusion

Too many conclusions were made. They should be further summarized to show the main conclusion directly connected to the main aim of this paper.

7. PLOS authors have the option to publish the peer review history of their article (what does this mean?). If published, this will include your full peer review and any attached files.

Reviewer #1: No

Reviewer #2: No

Reviewer #3: No

---

## [Author Response · Author response to Decision Letter 1]

6 Apr 2020

Dear Editors and Reviewers:

Thank you for your letter and for the reviewers’ comments concerning our manuscript entitled “Evaluation of Water Conservation Functions in the Xiongan New Area Based on a Comprehensive Index Method” (No.PONE-D-19-19149).Those comments are all valuable and very helpful for revising and improving our paper, as well as the important guiding significance to our researches. We have studied comments carefully and have made correction which we hope meet with approval. Revised portion are marked in red in the paper. The main corrections and responds to the reviewers’ comments are as follows.

Reviewer #1:

The authors have incorporated comments from the first round of review. My concerns from my previous review have been addressed. I would recommend the paper to be accepted for publication.

Response: Thanks to the recognition of the experts, we carefully read and revised our manuscript, hoping to reach the level of publication.

Reviewer #2:

1. What's the innovation of this paper? the significance? It's still not clear. The authors should summarized and showed it in a certain part in the paper, maybe in the end of the introduction.

Response: Thanks to the reviewer's advice. We have summarized the innovation of this paper in the end of the introduction according to the Reviewer’s suggestion.

2. The language should be more carefully corrected throughout the whole manuscript.

Response: Thanks to the reviewer's advice. We have corrected the language throughout the whole manuscript according to the Reviewer’s suggestion. In addition, the language of our manuscript have been refined and polished by a professional editing company.

3. Line 31-44, this paragraph may be shortened to just show the significance of this study. 

Response: Thanks to the reviewer's advice. We have shorted this paragraph according to the Reviewer’s suggestion and highlighted the importance and significance of water conservation function evaluation.

4. Line 45-63, the use of RS, the authors take so much sentences on the use of RS, Does it the main method or the features of this paper? What's more, so many literatures were just listed here. And it is the same with next paragraph. Another paragraph should be added to show how this paper to tackle this problems.

Response: Thanks to the reviewer's advice. We have made correction according to the Reviewer’s suggestion. (1) We originally intended to highlight the main research methods of ecological function evaluation, but now we feel that they do not have much relationship with the article, so we deleted line 45-63 directly according to the Reviewer’s opinion. (2) The next paragraph mainly describes the assessment methods of water conservation function and introduces a widely used model, InVEST. However, this method is not the method adopted in this paper, so the description of this paragraph is simplified according to the opinions of expert. (3) In the next paragraph, we added the application and advantages of the comprehensive index method used in this paper.

5. Line 120-121, Why data in April and May were chosen? and What about data in summer or the other season? and Why it wasn't in the same month?

Response: Thanks to the reviewer's advice. Due to the poor data quality of Landsat in May of 2005 in Xiongan New Area, the data of April which is the latest with the dates of 2010 and 2015 were originally selected. Although there is a certain deviation in time, the impact is not very big. Landsat data were used to calculate the TVDI index, which is jointly determined by NDVI and LST. The result shows that the drought was the severest in 2010, the least severe in 2015 and moderate in 2005. In addition, from the meteorological data analysis, the degree of drought difference in April and May is not big. The data of May are selected as the representative mainly because the Xiongan New Area is located in the north China plain, with a large amount of cultivated land, and winter wheat are mainly crops. May is the best period for the growth of winter wheat, which is conducive to the calculation of various indexes.

6. Line 130-131, What's the relation between the data of all the time nodes?

Response: Thanks to the reviewer's advice. The land cover data used in this study is the dataset of China's land use types released by the Chinese Academy of Sciences. The data were classified based on Landsat images. The late 1970s and late 1980s were built in intervals of 10 years, and were subsequently released every five years to analyze the change rules of China's land use.

7. Line 159, What's GI stand for?

Response: Thanks to the reviewer's advice. We have consulted a large number of literatures. Some people called this method "rank correction analysis method" and others "order relation analysis method". We referred to the opinions of most people and finally adopted "rank correction analysis method". "GI" is short for this method, but it is not authoritative, so we delete it this time.

8. Line 193, What's the difference between woodland and forestland? they both appeared in this paper.

Response: Thanks to the reviewer's advice. Due to our negligence, woodland and forestland both appeared in the article at the same time. We have carefully revised and changed all woodland into forestland.

9. Too many conclusions were made. They should be further summarized to show the main conclusion directly connected to the main aim of this paper.

Response: Thanks to the reviewer's advice. We have summarized the conclusion to highlight the aim of this paper according to the Reviewer’s suggestion.

---

## [Decision Letter · Decision Letter 2]

22 Apr 2020

PONE-D-19-19149R2

Evaluation of Water Conservation Function in the Xiongan New Area Based on the Comprehensive Index Method

PLOS ONE

Dear Mrs. Chen,

Thank you for submitting your manuscript to PLOS ONE. After careful consideration, we feel that it has merit but does not fully meet PLOS ONE’s publication criteria as it currently stands. Therefore, we invite you to submit a revised version of the manuscript that addresses the points raised during the review process.

We would appreciate receiving your revised manuscript by Jun 06 2020 11:59PM. To enhance the reproducibility of your results, we recommend that if applicable you deposit your laboratory protocols in protocols.io, where a protocol can be assigned its own identifier (DOI) such that it can be cited independently in the future. For instructions see: http://journals.plos.org/plosone/s/submission-guidelines#loc-laboratory-protocols

We look forward to receiving your revised manuscript.

Kind regards,

Rodolfo Nóbrega

Academic Editor

PLOS ONE

Reviewers' comments:

Reviewer's Responses to Questions

**Comments to the Author**

1. If the authors have adequately addressed your comments raised in a previous round of review and you feel that this manuscript is now acceptable for publication, you may indicate that here to bypass the “Comments to the Author” section, enter your conflict of interest statement in the “Confidential to Editor” section, and submit your "Accept" recommendation.

Reviewer #3: All comments have been addressed

2. Is the manuscript technically sound, and do the data support the conclusions?

Reviewer #3: Yes

3. Has the statistical analysis been performed appropriately and rigorously? 

Reviewer #3: Yes

4. Have the authors made all data underlying the findings in their manuscript fully available?

Reviewer #3: Yes

5. Is the manuscript presented in an intelligible fashion and written in standard English?

Reviewer #3: Yes

6. Review Comments to the Author

Reviewer #3: 1. “Compared with the InVEST model, the comprehensive index method is more convenient, and the parameters and weights can be adjusted according to the characteristics of the research area, which improves evaluation accuracy.”

How can this be proved？please explain more.

2. "2015" is missing in the third figure, which should be the same as the first two figures, such as Fig.3 and fig.8

7. PLOS authors have the option to publish the peer review history of their article (what does this mean?). If published, this will include your full peer review and any attached files.

Reviewer #3: No

---

## [Author Response · Author response to Decision Letter 2]

8 Aug 2020

1、Compared with the InVEST model, the comprehensive index method is more convenient, and the parameters and weights can be adjusted according to the characteristics of the research area, which improves evaluation accuracy.” How can this be proved？please explain more.

Response: Thanks to the reviewer's advice. There are many methods for evaluating water conservation functions, and each method has its advantages and limitations. We can't explain which method is better. We think that the expression of this paragraph is not very appropriate. We can only explain that the comprehensive index method used in this article is more convenient than the InVEST model. The comprehensive index method does not need localized calibration as the InVEST model. In addition, different index factors can be selected according to the characteristics of the study area. Thus, we revised the paragraph to “Each method has its advantages and limitations. In this paper, we choose the comprehensive index method to evaluate water conservation functions. This method more convenient than the InVEST model, which does not need localized calibration of model parameters, and the parameters can be adjusted according to the characteristics of the research area whenever necessary”

In addition, we have referred to a lot of literatures to supple the advantages and limitations of the InVEST model. Supplementary instructions are as follows:

The InVEST (Integrated Valuation of Ecosystem Services and Tradeoffs) model was jointly developed by Stanford University, the Nature Conservancy and the World Wide Fund for Nature to assess ecosystem service functions. InVEST model has multiple modules, covering almost all ecological service functions. The water conservation module is an important part of the InVEST model, which is based on the water balance method. InVEST model has been successfully applied to Willamette River in the USA[1], Cote d'Ivoire in West Africa[2], Francoli basin in northeastern Spain[3]. Chinese scholars successfully applied the InVEST model to Beijing [4], Dujiangyan[5], Sanjiangyuan District[6,7], Qinling Mountains[8], Fujian[9], Dongting Lake[10] and other regions in China. The applicability of InVEST model and the local suitability of parameters are the key to the reliability of the model. However, the calculation methods and parameters in the InVEST model are mostly based on American standards. When applied to the regions of China, it is necessary to establish a constantly improving database that conforms to the features of regional ecosystem service functions, and use the measured data to correct the model parameters. In addition, data acquisition is not easy and is sensitive to data changes. Furthermore, the InVEST model takes the land use type as a unit, and does not consider the topography and other factors.

[1] Erik Nelson, Guillermo Mendoza, James Regetz, Stephen Polasky, Heather Tallis, DRichard Cameron, et al. Modeling multiple ecosystem services, biodiversity conservation, commodity production, and tradeoffs at landscape scales. Frontiers in Ecology and the Environment.2009; 7(1):4-11.

[2] Leh, M.D.K., Matlock, M.D., Cummings, E.C., Nalley, L.L. Quantifying and mapping multiple ecosystem services change in West Africa. Agriculture, Ecosystems and Environment. 2013; 165, 6-18.

[3] Marquès M, Bangash RF, Kumar V, Sharp R, Schuhmacher M. The impact of climate change on water provision under a low flow regime: a case study of the ecosystems services in the Francoli river Basin. Journal of Hazardous Materials.2013; 263:224-32.

[4] Yu XX, Zhou B, Lv XZ, Yang ZG. Evaluation of Water Conservation Function in Mountain Forest Areas of Beijing Based on InVEST Model. Scientia Silvae Sinicae. 2012; 48(10):1-5.

[5] Fu B, Xu P, Wang YK, Peng Y, Ren J. Spatial Pattern of Water Retetnion in Dujiangyan County. Acta Ecologica Sinica. 2013; 33(3):0789-97.

[6] Pan T, Wu SH, Dai EF, Liu YJ. Spatiotemporal variation of water source supply service in Three Rivers Source Area of China based on InVEST model. Chinese Journal of Applied Ecology.2013;24(1):183-9.

[7] Lv LT, Ren TT, Sun CZ, Zheng DF, Wang H. Spatial and temporal changes of water supply and water conservation function in Sanjiangyuan National Park from 1980 to 2016. Acta Ecologica Sinica. 2020; 40(3):993-1003.

[8] Fan YN, Liu K, Chen SS, Yuan J G. Spatial pattern analysis on water conservation functionality of land ecosystem in Northern slope of Qinling Mountains. Bulletin of Soil and Water Conservation. 2017; 37(2): 50-6.

[9] Wang BS, Chen HX, Dong Z, Zhu W, Qiu QY, Tang LN. Impact of land use change on the water conservation service of ecosystems in the urban agglomeration of the Golden Triangle of Southern Fujian, China, in 2030. Acta Ecologica Sinica. 2020;40(2) :484-98.

[10] Hu WM, Li G, Gao ZH, Jia GY, Wang ZC, Li Y. Assessment of the impact of the Poplar Ecological Retreat Project on water conservation in the Dongting Lake wetland region using the InVEST model. Science of The Total Environment. 2020.

2. "2015" is missing in the third figure, which should be the same as the first two figures, such as Fig.3 and fig.8

Response: Thanks to the reviewer's advice. Due to our negligence, "2015" was missing in some figures. We have checked and revised all figures carefully.

---

## [Editor Report · Decision Letter 3]

25 Aug 2020

Evaluation of Water Conservation Function in the Xiongan New Area Based on the Comprehensive Index Method

PONE-D-19-19149R3

Dear Dr. Chen,

We’re pleased to inform you that your manuscript has been judged scientifically suitable for publication and will be formally accepted for publication once it meets all outstanding technical requirements.

Kind regards,

Rodolfo Nóbrega

Academic Editor

PLOS ONE

Additional Editor Comments (optional):

Please make sure you read the entire manuscript again to eliminate some typos and grammatical mistakes before the submission of the final manuscript version.
---

## [Editor Report · Acceptance letter]

31 Aug 2020

PONE-D-19-19149R3 

Evaluation of Water Conservation Function in the Xiongan New Area Based on the Comprehensive Index Method 

Dear Dr. Chen:

I'm pleased to inform you that your manuscript has been deemed suitable for publication in PLOS ONE. Congratulations! Your manuscript is now with our production department. 

Kind regards, 

on behalf of

Dr. Rodolfo Nóbrega 

Academic Editor

PLOS ONE